# Uncertainty Quantification from
# Calcified Plaque to Outer Wall Estimation in CTA

**Jiehyun Kim**[1] [ID]                                        JIEHYUN.KIM001@UMB.EDU
[1] *University of Massachusetts Boston, Boston, MA, USA*
**Huy Q. Phi**[2]                                        HUY.PHI@PENNMEDICINE.UPENN.EDU
[2] *Drexel University College of Medicine, Philadelphia, PA, USA*
**Grace J. Wang**[3]                                    GRACE.WANG@PENNMEDICINE.UPENN.EDU
[3] *University of Pennsylvania, Philadelphia, PA, USA*
**Brett L. Cucchiara**[3]                                CUCCHIAR@PENNMEDICINE.UPENN.EDU
**Elias Johansson**[4]                                    ELIAS.JOHANSSON@NEURO.GU.SE
[4] *University of Gothenburg, Gothenburg, Sweden*
**Jae W. Song**[3]                                        JAE.SONG@PENNMEDICINE.UPENN.EDU
**Daniel Haehn**[1]                                        DANIEL.HAEHN@UMB.EDU

## Abstract

Deep learning methods have shown strong performance for calcified plaque segmentation on CTA. However, the reliability of voxel-level predictions remains underexplored. We present an nnUNet-based framework that estimates voxel-wise uncertainty from a 5-fold ensemble. Rejecting low-confidence voxels improves both segmentation accuracy and calibration. We further extend the analysis to outer vessel wall estimation using variance-based tissue characterization, where ground-truth annotation is difficult and inconsistent. Our framework achieves an AUROC of 0.954 for voxel-level error detection and provides an integrated visualization of plaque hounsfield units (HU) values, prediction probability, uncertainty, and outer-wall variance. These results demonstrate that uncertainty-aware analysis supports both reliable plaque segmentation and exploratory vessel wall characterization. All implementations are open source on our GitHub repository: https://github.com/mpsych/CACTAS-UQ.

**Keywords:** Calcified Plaque, Outer Wall Estimation, CTA, Uncertainty Quantification

## 1. Introduction

Deep learning methods have shown promising performance for carotid artery and plaque segmentation on CTA (Zhai et al., 2024; Luo et al., 2025; Owen et al., 2011). However, segmentation accuracy alone does not capture voxel-level reliability, particularly near plaque boundaries where tissue contrast is ambiguous. Uncertainty quantification (UQ) can identify the calibration of automated outputs. Beyond plaque segmentation, the outer vessel wall is also relevant to carotid plaque assessment for measuring total plaque thickness. However, it is challenging due to low contrast and inconsistent annotations (Schlett et al., 2013). In this setting, the challenge is not only model uncertainty, but also uncertainty in the underlying label definition itself. As a result, standard supervised segmentation is difficult to apply reliably for outer wall analysis.

In this work, we combine ensemble-based uncertainty quantification for calcified plaque segmentation with variance-based outer wall estimation where reliable annotation is unavailable. We further combine plaque Hounsfield Unit (HU) values, prediction probabilities, uncertainty maps, and variance maps into an integrated visualization that links plaque composition and model reliability.

## 2. Methods

**Data** We analyzed 70 CTA volumes from patients the Penn Stroke Registry with expert annotations of calcified plaque. The Dataset was split into 56 training and 14 test cases. Lumen annotations were available for a subset of 27 cases used for wall characterization.

**Calcified Plaque Segmentation** We used nnU-Net with a residual encoder in 3D full-resolution configuration, trained with 5-fold cross-validation (Isensee et al., 2021, 2024) (See Appendix Table 1). Carotid artery masks were applied to focus input on the vessel region (Kim et al., 2025).

**Ensemble Uncertainty Quantification** To estimate voxel-wise predictive uncertainty, we used the outputs of the five cross-validation models as an ensemble (Huang et al., 2024). Five uncertainty metrics were evaluated: standard deviation, variance, range, disagreement rate, and mutual information. Rejection thresholds were selected on training data and fixed for test evaluation. The evaluation region was defined by a 3-voxel dilation of the ground truth.

**Evaluation Metrics** Uncertainty quality is assessed by AUROC, where higher AUROC indicates better separation between correctly classified and misclassified voxels. Segmentation quality before and after rejection is measured by Dice, intersection over uniton (IoU), expected calibration error (ECE), and Brier score.

**Outer Wall Estimation** For 27 cases with lumen annotations, the outer wall region is estimated by merging lumen and plaque masks, resampling to 0.5 mm isotropic spacing, filling holes, dilating by 5 voxels ($\approx$2.5 mm), and resampling back. The wall region is defined as the dilated area minus the lumen-plaque union. A local HU variance map was computed, and Otsu thresholding was applied to distinguish heterogeneous vessel wall tissue from more homogeneous surroundings. This provides a data-driven outer wall estimate without manual annotations.

## 3. Results and Discussion

**Segmentation and Uncertainty Quantification of Calcified Plaque** The baseline segmentation achieved a Dice of 0.8322 (Appendix Table 1). Among five uncertainty metrics, standard deviation and variance performed best for error detection (AUROC=0.9543), while disagreement rate performed worst (AUROC=0.7104) (Appendix Table 2, Figure 3). Standard deviation was selected for further analysis. Rejecting voxels above threshold $t = 0.01$ improved Dice to 0.8921 and reduced both ECE and Brier score to 0.0103. Uncertainty concentrated at plaque boundaries where errors were most common, and rejection produced cleaner segmentation boundaries consistent with the improved metrics (Figure. 1). The integrated visualization presents plaque HU composition, model probability, and un-

certainty in a single view, showing that high-uncertainty regions align with boundary areas where model confidence is lowest (Figure 2).

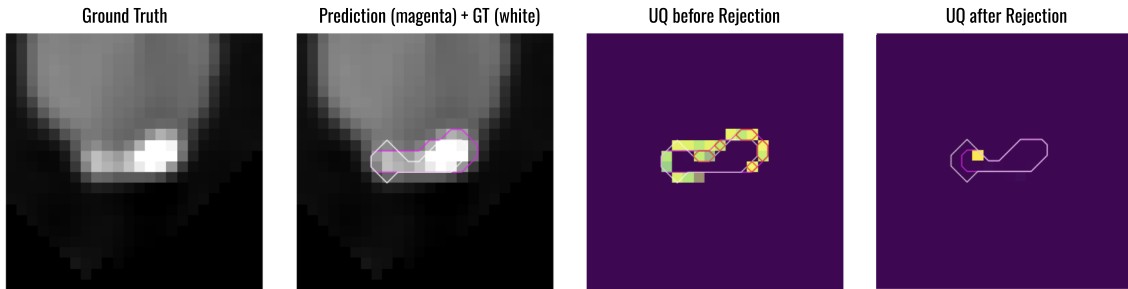

Figure 1: High uncertainty appears at boundaries, and removing these regions yields cleaner segmentation and improved calibration, demonstrating that uncertainty highlights unreliable predictions.

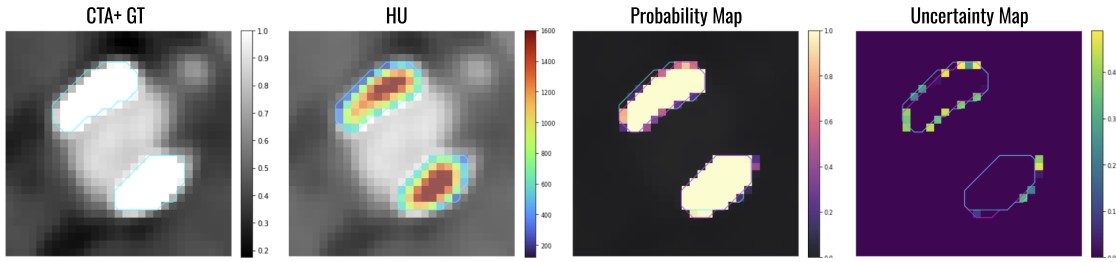

Figure 2: Shows plaque composition, model confidence gradient, and boundary uncertainty in a single view. Uncertainty concentrates at plaque boundaries where the model is least confident.

**Outer Wall Estimation** Variance-based analysis revealed a high-variance band surrounding the lumen-plaque complex, consistent with the vessel wall. Outside this band, variance decreased sharply, suggesting a transition to more homogeneous surrounding tissue. Otsu thresholding enabled adaptive detection of this boundary. In plaque-bearing cases, higher and more heterogeneous variance was observed near plaque, suggesting increased local tissue complexity (Appendix Figure. 4). This outer-wall analysis suggests that local variance may provide useful structural information in CTA when reliable manual labels are unavailable.

**Limitation** The dataset is relatively small(n=70, 14 test) which may limit generalizability. Second, analyses are restricted to calcified plaque and do not address non-calcified plaque. Third, the outer-wall component is exploratory and limited to 27 cases with lumen annotations.

## 4. Conclusion

Ensemble-based voxel-wise uncertainty improves the reliability of calcified plaque segmentation in CTA, providing information beyond the segmentation mask. Future work will include outer-wall ground truth to segment total plaque (calcified + non-calcified plaque) and multi-center validation.

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

## Appendix A. nnUNet Segmentation

Table 1: Plaque segmentation performance metrics (masked versions). Higher values indicate better performance.

| Model | Type | IoU | Dice |
|---|---|---|---|
| 2D nnUNet | Baseline | 0.5678 | 0.7111 |
| 2D nnUNet | Residual (R+p+b) | 0.5652 | 0.6958 |
| 3D nnUNet | Baseline | 0.6806 | 0.8051 |
| 3D nnUNet | Residual (R+p+b) | 0.7184 | 0.8322 |

R+p+b: residual encoder with patch and batch adjustments.

## Appendix B. Uncertainty Quantification Results

Table 2: Uncertainty quantification results on the test set (n=14) using 5-fold ensemble predictions. Baseline (shared across methods): IoU=0.7263, Dice=0.8379, ECE=0.0165, Brier=0.0180. Values after rejection at method-specific thresholds selected on training data.

| Method | IoU↑ | Dice↑ | ECE↓ | Brier↓ | AUROC↑ | Threshold |
|---|---|---|---|---|---|---|
| std | **0.8121** | **0.8921** | **0.0103** | **0.0103** | **0.9543** | 0.01 |
| var | 0.7922 | 0.8789 | 0.0118 | 0.0119 | **0.9543** | 0.01 |
| range | 0.8110 | 0.8915 | 0.0104 | 0.0104 | 0.9504 | 0.03 |
| disagree | 0.7504 | 0.8531 | 0.0151 | 0.0156 | 0.7104 | 0.20 |
| MI | 0.8110 | 0.8915 | 0.0104 | 0.0105 | 0.8628 | 0.01 |

## Appendix C. Paired t-test

Table 3: Statistical significance analysis between baseline and uncertainty-based rejection results using paired t-test (n=14).

| Metric | t-statistic | p-value |
|---|---|---|
| IoU | -6.27 | < 0.0001 |
| Dice | -6.03 | < 0.0001 |
| ECE | 7.28 | < 0.0001 |

## Appendix D. AUROC among five UQ methods

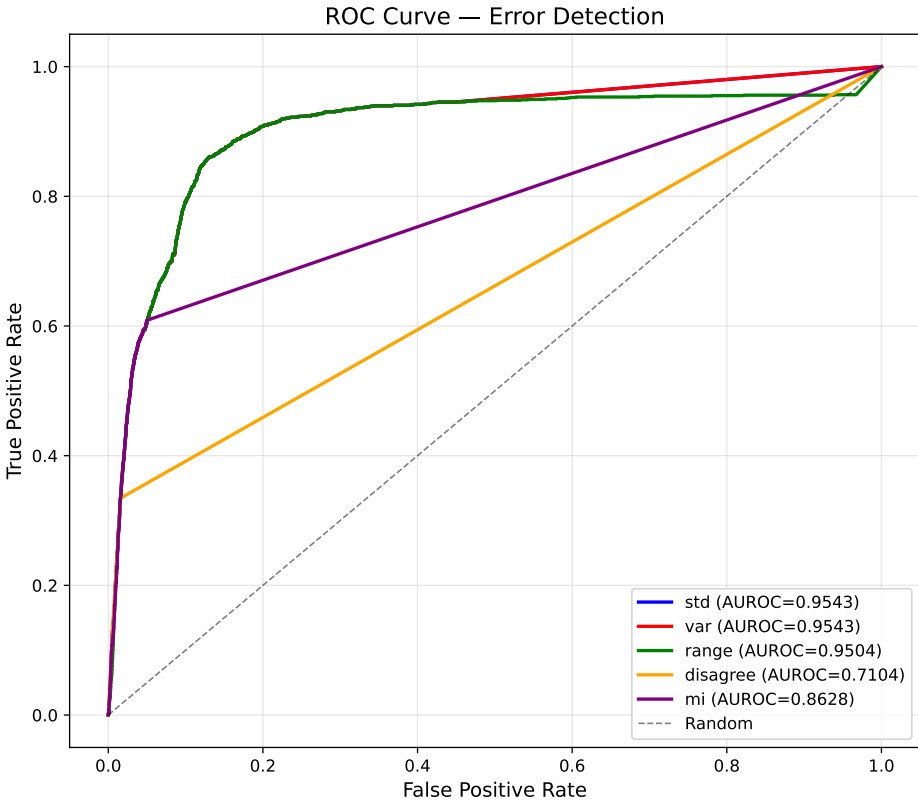

Figure 3: ROC curves for error detection across five uncertainty metrics. Standard deviation and variance achieve the highest AUROC (0.954), while disagreement rate performs worst (0.710). The dashed line indicates random performance.

## Appendix E. Outer Wall Estimation with and without plaque

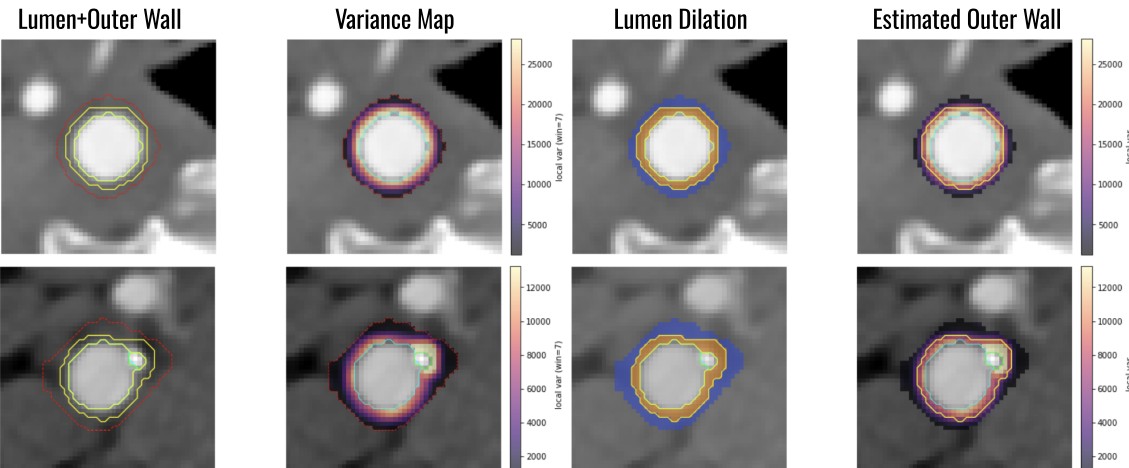

Figure 4: Outer wall estimation for two representative cases: with plaque (top) and without plaque (bottom). From left to right: CTA with anatomical contours (lumen in cyan, plaque in lime, dilation boundary in red, Otsu boundary in yellow); local variance heatmap on the wall region; Otsu segmentation (wall tissue in orange, outside in blue); variance map with Otsu boundary overlay. The plaque side shows higher and more heterogeneous variance compared to the plaque-free side.

