# OpenReview forum: "Uncertainty Quantification from Calcified Plaque to Outer Wall Estimation in CTA"
_MIDL.io/2026/Short_Papers — MIDL 2026 - Short Papers Poster_

### Official Review · Reviewer_5jdi · 2026-05-02
**ensemble-based uncertainty quantification (UQ) framework for calcified plaque segmentation in carotid CT angiography (CTA) images**

**Rating:** 5
**Confidence:** 4

**Review:**

This paper adresses the important problem of uncertainty quantification for calcified plaque segmentation in carotid CT angiography (CTA) images. The authors propose an ensemble-based uncertainty quantification (UQ) framework. Segmentation is performed by nnU-Net and voxel-level uncertainty is derived from the ensemble probability maps. Proposed framework extends from reliable plaque boundary detection to exploratory outer wall estimation using variance-based tissue characterization.
The paper is clear and well-written. All implementations are open source on their GitHub repository.

**Summary:**

This paper presents an ensemble-based uncertainty quantification (UQ) framework for calcified plaque segmentation in carotid CT angiography (CTA) images. Segmentation is performed by nnU-Net and voxel-level uncertainty is derived from the ensemble probability maps. Proposed framework extends from reliable plaque boundary detection to exploratory outer wall estimation using variance-based tissue characterization.

**Strengths:**

This paper adresses the important problem of uncertainty quantification for calcified plaque segmentation in carotid CT angiography (CTA) images. The authors combine ensemble-based uncertainty quantification for calcified plaque segmentation with variance-based outer wall estimation where reliable annotation is unavailable.
The paper is clear and well-written.
All implementations are open source on their GitHub repository.

**Weaknesses:**

As indicated by the authors, the dataset is small (n=70, 14 test) and comes from only one center which may limit generalizability.
Moreover it could have been interesting to adress also non-calcified plaque.
The outer-wall component is limited to 27 cases with lumen annotations.

**Justification Of Rating:**

This paper adresses the important problem of uncertainty quantification for calcified plaque segmentation in carotid CT angiography (CTA) images. The authors combine ensemble-based uncertainty quantification for calcified plaque segmentation with variance-based outer wall estimation where reliable annotation is unavailable.
The paper is clear and well-written.
All implementations are open source

---

### Decision · Program_Chairs · 2026-05-08

Accept (Poster)